# Multidimensional Disadvantages of a Gluten-Free Diet in Celiac Disease: A Narrative Review

**DOI:** 10.3390/nu13020643

**Published:** 2021-02-16

**Authors:** Martyna Marciniak, Aleksandra Szymczak-Tomczak, Dagmara Mahadea, Piotr Eder, Agnieszka Dobrowolska, Iwona Krela-Kaźmierczak

**Affiliations:** Department of Gastroenterology, Dietetics and Internal Diseases, Poznan University of Medical Sciences, 49 Przybyszewskiego Street, 60-355 Poznan, Poland; aleksandra.szymczak@o2.pl (A.S.-T.); piotreder@ump.edu.pl (P.E.); agdob@ump.edu.pl (A.D.); krela@op.pl (I.K.-K.)

**Keywords:** celiac disease, gluten-free diet, dyslipidemia, obesity, diabetes mellitus t.2, metabolic syndrome, cardiovascular diseases, gut microbiota, microbiome, gut dysbiosis

## Abstract

A gluten-free diet is the mainstay method of treatment and the prevention of celiac disease complications. However, an inadequately balanced gluten-free diet can increase the risk of obesity, negatively affect glucose and lipid metabolism, and increase the risk of the metabolic syndrome. Therefore, an adequate nutritional counselling is necessary for patients diagnosed with celiac disease in order to prevent and treat the components of the metabolic syndrome.

## 1. Introduction

Celiac disease (CD) is a chronic autoimmune enteropathy of the small intestine which lasts a lifetime [1,2,3]. It is characterized by digestive, as well as absorption disorders, and is caused by an abnormal immune response to prolamines (alcohol-soluble protein fractions) contained in cereals [4]. These compounds, commonly known as gluten, include wheat protein, gliadin; the protein contained in rye, secalin; and in barley, hordein [5,6]. According to the hypothesis of Kӧttgen, the occurrence of celiac disease is determined by genetic, environmental, infectious, metabolic, and immunological factors [7,8,9,10,11].

In the general population, the prevalence of celiac disease amounts to 0.5–1% [12,13]. An increased occurrence of celiac disease can be observed in individuals with autoimmune diseases, such as type 1 diabetes, autoimmune hepatitis, thyroiditis, alopecia areata, *Dermatitis Herpetiformis,* or psoriasis [13,14,15,16]. While the incidence of celiac disease in autoimmune hepatitis ranges between 3–6% in adults and about 20% in children, the co-occurrence of type 1 diabetes and CD can be found in 1–10% of patients in the European population [17,18]. An increased risk of developing celiac disease has been reported in patients affected by the chromosomal disorders, such as Down’s syndrome, as well as Williams’ syndrome, and Turner’s syndrome [19,20]. According to Fasano et al., the prevalence of CD amounted to 1:10 in first-degree relatives and 1:39 in second-degree relatives [21]. A systematic review conducted in the MEDLINE (1966–2003) and EMBASE (1947–2003) databases highlights the incidence of celiac disease in 20% of first-degree relatives [22]. A gluten-free diet (GFD) is the mainstay method of both treatment and the prevention of celiac disease complications. However, it should be emphasized that an inadequately balanced gluten-free diet, rather than CD diagnosis, may result in the metabolic consequences [23]. Gluten elimination is recommended only in the treatment of clinical gluten sensitivity, e.g., in celiac disease, or non-celiac gluten sensitivity. [24] A randomized, double-blind trial involving 30 volunteers who had not been diagnosed with celiac disease, did not reveal any symptomatic benefits of a GFD in individuals without gluten-dependent diseases. DRCT (double-blind randomized controlled trial) has also demonstrated that gluten consumption does not cause any symptoms in healthy volunteers [25].

A gluten-free diet requires a complete elimination of wheat, rye and barley [26,27,28]. Indeed, even as little as 50 mg of gluten, which can be found in a few breadcrumbs or a small piece of pasta, can increase enteropathy [29]. Furthermore, many products can be contaminated with gluten during harvesting, processing or packaging [27]. A good example can be gluten-free oats, which are often contaminated with wheat or barley [30]. Furthermore, wheat is often used as a thickening or filling agent for numerous meat products, soups, sauces, and as an coating for meat or eggs [31]. According to the Codex Alimentarius Commission, gluten-free products are foods which contain ≤20 ppm gluten (20 mg of gluten per 1 kg of food sold) [32]. A growing body of evidence suggests that most individuals with celiac disease will tolerate foods with low gluten content (≤20 mg/kg) [33]. Nevertheless, patients with non-Responsive Celiac Disease (NRCD) are particularly sensitive even to trace amounts of gluten [34]. In the study of Hollon JR et al., patients with suspected refractory celiac disease type 1 followed a diet eliminating gluten completely for a period of 3–6 months. This alimentation, which was solely based on fresh, unprocessed foods, was defined as the Gluten Contamination Elimination Diet (GCED). Consequently, a remission of some symptoms was observed in the patients with NRCD taking part in the study [35]. It is believed that GCED can also be used to distinguish between individuals who are “hypersensitive” to the classic gluten-free diet and patients with true Refractory Celiac Disease (RCD) [36]. According to the study conducted by Freeman which included 182 patients, a regeneration and normalization of the small intestinal mucosa could occur even within 6 months of a strict compliance to a gluten-free diet, while the best results were observed after a year or two of such a procedure. Furthermore, it was observed that after the six-month-long exclusion of gluten, the health of women improved to a greater extent than that of men. However, in individuals over 65 years of age, the improvement was slower than in younger age groups, although it was mainly observed in men [37].

A gluten-free diet is the only effective method of treating celiac disease. However, it may increase the risk of obesity, themetabolic syndrome, as well as negatively affect glucose,lipid metabolism and the intestinal microbiota, it is shown in Figure 1. This paper reviews the available published papers regarding this topic.

## 2. Material and Methods

In order to collect the literature data related to the presented topic, the PubMed database (www.pubmed.ncbi.nlm.nih.gov, accessed on 22 April 2020) was explored using the terms: “celiac disease”, “gluten-free diet”, “dyslipidemia”, “obesity”, “diabetes mellitus t.2”, “metabolic syndrome”, “cardiovascular diseases”, “gut microbiota”, and “microbiome gut dysbiosis”.

## 3. GFD and Its Risks

### 3.1. Gluten-Free Diet and Obesity

Although one of the classic manifestations of celiac disease is body weight deficiency, the number of patients suffering from overweight or obesity at the time of diagnosis is on the rise [38,39,40,41,42,43]. Research conducted over the past decade suggested that following a gluten-free diet may result in weight gain [44,45,46]. In fact, Valletta et al. demonstrated a significant increase in body weight following the implementation of a gluten-free diet [47]. Additionally, the researchers hypothesized that the reduction in dietary fiber, excess fats and the supply of hypercaloric drinks and high-glycemic grain products resulted in an elevated body weight [47,48,49,50,51,52]. Moreover, the authors emphasized that the improper eating habits of these patients may be due to the reduced taste values of gluten-free products, which in turn translated into an increased consumption of sweet and salty snacks [44]. This hypothesis is supported in the study conducted by Dall’Asta C. et al., which indicated a preference in high-fat meals and a high consumption of sweets as well as sweetened, colorful drinks, and a low consumption of fruit or vegetables among the 118 study participants [53]. Furthermore, two meta-analyses from 2018 and 2019 demonstrated that both adults and children on a gluten-free diet consumed too much energy, fats, cereals with a high glycemic index and a low content of dietary fiber and nutrients [54,55].

In 1986, Semeraro L.A. suggested that an increase in the absorption in the distal sections of the intestine might compensate the jejunal atrophy in patients with celiac disease [56]. He assumed that this process might be similar to structural changes in the residual intestine occurring after a partial surgical resection of the intestine. This adaptation includes morphological changes, such as an increase in villi height, as well as an increase of the depth of the intestinal crypts, and the number of intestinal epithelial cells. Furthermore, in patients with CD, atrophy causes loss of normal bowel function which may induce an increased absorption in the functionally preserved segments of the gastrointestinal tract. Thus, if this process leads to an overcompensation, it may result in the child’s energy requirements being exceeded, thus increasing the risk of overweight or obesity [56]. This compensatory hypothesis seems to be supported by some of the first published cases of adolescents with CD who were still overweight or obese despite villi atrophy in the jejunal biopsy [57,58]. The compensatory surface area of the small intestine appears to increase with age. Therefore, the intestines can develop the ability to absorb the adequate amount of energy [56]. This is confirmed by the broad spectrum of symptoms following the diagnosis of CD, which appears to be age-related [59,60,61]. Moreover, while children under 2 years of age often present with the classic form of CD, including malabsorption, older children, adolescents, and adults report unusual symptoms [3,62] whichseems to be consistent with the compensation hypothesis. In fact, the classic symptoms may be due to a lack of intestinal adaptation, which is less developed in the young individuals. Additionally, a lack of intestinal adaptation causes severe and classic symptoms, including malabsorption and visceral crisis, which occur in very young children recently diagnosed with CD. Intestinal adaptation is a time-dependent phenomenon, and the likelihood of modifying a person’s small intestine mucosa increases with age; hence, it is possible to relieve the symptoms of CD in older children and adolescents [63]. According to this hypothesis, there is no correlation between CD presentation and the degree of villous atrophy or the degree of the intestinal involvement visualized by endoscopic procedures and video-capsules [64,65]. Moreover, the nutritional status of the general population is of utmost importance for the correct interpretation of BMI in children at the time of the diagnosis of CD. It is vital to bear in mind that celiac disease can develop in overweight/obese patients, reflecting the individual predispositions (i.e., genetic, nutritional and environmental factors) [66,67]. Furthermore, the worldwide prevalence of overweight and obesity in children has increased over the past two decades; in fact, it is estimated that 60 million children will have been overweight or obese by 2020 [68]. In a 10-year study by Dickey W. involving 371 patients, the mean BMI amounted to 24.6 kg/m2 (range 16.3–43.5). 17 patients (5%) were underweight (BMI <18.5), 211 (57%) had a normal body weight and 143 (39%) were overweight (BMI ≥ 25), of whom 48 (13% of all patients) were obese (BMI ≥ 30.0). In the groups of patients following a gluten-free diet, 81% gained weight after 2 years, with 82% of subjects initially being overweight [69]. On the other hand, Murray et al. demonstrated that 6% of 215 adult CD patients who were obese at the time of the diagnosis, presented with a decrease in BMI following 6 months of a GFD [70]. According to West et al., 3.9% of 3590 CD patients were obese, and 17% were overweight [71]. Furthermore, Viljamaa et al., in their 14-year study involving 50 patients with CD, found that 15% of them were categorized as obese [72]. Conversely, in the prospective study involving 698 subjects with newly diagnosed celiac disease, 4% of the participants were underweight, 57% presented a normal body weight, 28% were overweight, and 11% were obese. Following the implementation of a gluten-free diet, 69% of the weight-deficient patients gained weight, and 18% of those who were overweight and 42% of the obese patients experienced weight loss [73]. A comparison of mean BMI values at the time of the diagnosis and after following a gluten-free diet is shown in Table 1.

### 3.2. Gluten-Free Diet and the Metabolic Syndrome

A gluten-free diet is based on the products which have a high glycemic index and are devoid of dietary fiber. These foods contain many simple carbohydrates and fats. These factors can give rise to nutritional deficiencies, constipation, and the development of the metabolic syndrome [82]. The malabsorption syndrome observed in celiac disease, as well as an improperly balanced gluten-free diet lead to nutritional deficiencies including iron, folic acid, calcium, and vitamin D, as well as B vitamins and zinc [83].

Numerous publications emphasize that an improperly balanced gluten-free diet may entail the development of the metabolic syndrome due to the consumption of high-glycemic cereal products and foods with a high content of fatty acids [74,79,84]. The metabolic syndrome is referred to as a set of interrelated factors which significantly increase the risk of developing atherosclerosis, type 2 diabetes, and their cardiovascular complications. The consensus of the International Diabetes Federation (IDF), National Heart, Lung, and Blood Institute (NHLBI), American Heart Association (AHA), World Heart Federation (WHF), International Atherosclerosis Society (IAS), and the International Association for the Study of Obesity (IASO) dating back to 2009 states that for the diagnosis of the metabolic syndrome (MetS) any three of the following five criteria should be fulfilled:  Waist circumference (depending on the country of origin and ethnic group—in the European population ≥80 cm in women and ≥94 cm in men);  Triglyceride concentration>1.7 mmol/L (150 mg/dL) or the treatment of hypertriglyceridemia;  HDL-C (high-density lipoprotein) concentration <1.0 mmol/L (40 mg/dL) in men and <1.3 mmol/L (50 mg/dL) in women or the treatment of low HDL-C;  Systolic blood pressure ≥130 mmHg or diastolic blood pressure ≥85 mmHg, or the treatment of previously diagnosed arterial hypertension;  Fasting glucose ≥ 5.6 mmol/L (100 mg/dL) or a drug treatment of type 2 diabetes [85].

In the study of Tortora R involving 98 newly diagnosed patients, it was observed that the number of individuals diagnosed with the metabolic syndrome 6 vs. 29 (*p* < 0.01; OR: 20) was increased following one year of a GFD. Pertaining to the diagnosis of the metabolic syndrome, a rise was observed in the number of participants presenting with the following: an increased waist circumference 48 vs. 72 (*p* < 0.01; OR: 2.8), high blood pressure 4 vs. 18 (*p* < 0.01; OR: 5.2), exceeded glucose threshold 7 vs. 25 (*p* = 0.01; OR: 4.4) and an elevated triglycerides concentration level 7 vs. 16 (*p* = 0.05) [74]. Moreover, similar conclusions were reached by Italian researchers who enrolled 185 adults with celiac disease in their study. At the time of the diagnosis, the metabolic syndrome was diagnosed in 3.24% (*n* = 6) of the participants, and after the introduction of a gluten-free diet, the number of patients with MetS increased to 14.59% (*n* = 26; *p* < 0.0001) [79]. A prospective study, including 301 patients with celiac disease, evaluated the predictors of the metabolic syndrome at the time of the diagnosis and after one year of gluten-free dieting. While upon the diagnosis 4.3% of the participants met the criteria for MetS, 23.9% developed the metabolic syndrome (4.3% vs. 23.9%; *p* < 0.001; OR 6.9) following 1 year of a GFD [84]. In the retrospective study conducted by Kabbani et al. 840 patients with celiac disease were compared with 840 healthy individuals in terms of age, gender, and ethnicity. The incidence of the metabolic syndrome was significantly lower in patients with celiac disease, as compared to the control group (3.5% vs. 12.7%; *p* < 0.0001). In fact, the mean BMI of patients with CD was significantly lower than in the control group (24.7 vs. 27.5; *p* < 0.0001) [86].

### 3.3. Gluten-Free Diet and Dyslipidemia

An inadequately balanced gluten-free diet may be associated with an increase in the total cholesterol, LDL, triglycerides, and a decreased concentration of HDL fraction [79,81,87]. In the following research studies, lipids concentrations were evaluated before and after the introduction of a gluten-free diet. Tortora R et al. compared the concentration of triglycerides at the time of the diagnosis and after 4 years of a gluten-free diet compliance. However, in another prospective study, the same authors did not demonstrate statistically significant changes in the triglyceride levels following a year of a gluten-free diet [74,75]. A cohort retrospective study involving 185 celiac disease patients revealed an increase in cholesterol and triglycerides, as well as a reduction in high-density lipoproteins after complying to a gluten-free diet [79]. On the other hand, in one cross-sectional study, the incidence of dyslipidemia was significantly lower in comparison to the control group (18.3% vs. 34.9%; *p* < 0.0001) [86]. The lipid profile of patients at the time of the CD diagnosis and after the introduction of a gluten-free diet is shown in Table 2.

### 3.4. Gluten-Free Diet and Cardiovascular Diseases (CVD)

The impact of a gluten-free diet on the development of cardiovascular disease remains unclear. There are studies which confirm the atherogenic effects of a gluten-free diet, whereas others report that this diet may have antiatherosclerosis effects. In the study by Brar P et al. involving 132 patients with celiac disease, the concentration of total cholesterol, LDL, and HDL fractions was assessed after six months of adherence to a strict gluten-free diet. The exclusion criteria comprised an increased concentration of total cholesterol, LDL fraction, triglycerides, diabetes, thyroid, liver, and pancreatic diseases at the time of the diagnosis of CD. An increase in the total cholesterol and high-density lipoprotein (*p* < 0.0001), but not the low-density lipoprotein (*p* = 0.06) was observed. The study indicated that the LDL/HDL ratio in the study group decreased by 0.36 ± 0.7 (*p* < 0.0001) [64]. In contrast, Zanini B. et al. in the retrospective analysis of the effect of a 1–5 years of a gluten-free diet with715 celiac patients, demonstrated a significant increase in BMI (21.4 ± 3.4 vs. 21.4 ± 3.4 vs. 22.5 ± 3.5; *p* < 0.0001), in the total cholesterol (171.2 ± 37.4 mg/dL vs. 181.4 ± 35.1 mg/dL; *p* < 0.0001), and in γ-Glutamyl transpeptidase (16.5 ± 14.9 vs. 19.5 ± 19.2 U/L; *p* < 0.0001), with a simultaneous reduction in the concentration of triglycerides (87.9 ± 49.5 vs. 80.2 ± 42.8 mg/dL; *p* < 0.0001) and homocysteine (16.9 ± 9, 6 vs. 13.3 ± 8.0 μmol/L; *p* = 0.018). Nevertheless, in this case, due to the unavailability of comprehensive information, it was not possible to estimate the CVD risk. Therefore, an alternative concept of patients with “a low risk of CVD” was applied which included a combination of factors, such as BMI> 25, blood glucose <100 mg/dL, total cholesterol <200 mg/dL, triglycerides <150 mg/dL and γ -GT 14–27 U/L, which decreased from 58% to 47% [80]. A cross-sectional study conducted at the Israel Schneider Medical Center for Children (Petach Tiqva, Israel) and San Paolo Hospital (Milan, Italy) enrolled 114 children with CD in a serologic remission who had been on a gluten-free for at least one year. This study evaluated BMI, waist circumference, LDL, triglycerides, blood pressure, and insulin resistance. The authors found three or more coexisting CVD factors in 14.4% of the subjects, where the most common ones included a high level of triglycerides (34.8%), increased blood pressure (29.4%), and a high LDL cholesterol (24.1%) [88].

### 3.5. Celiac Disease, a Gluten-Free Diet, and Type 2 Diabetes

The data regarding the association of celiac disease, a gluten-free diet, and the development of type 2 diabetes is inconsistent. Despite the fact that extensive evidence reports that CD patients are less likely to develop carbohydrate disorders, numerous studies correlate the use of a highly processed gluten-free diet with an increased risk of elevated glucose levels [74,80,86]. A cross-sectional study involving 840 patients with celiac disease and 840 controls matched by age, gender and ethnicity, revealed that subjects with CD presented a three times lower incidence of DM2 compared to healthy controls (26 vs. 81). However, it should be noted that one of the alternative possibilities is the overlapping of genes which predispose to celiac disease and protect against type 2 diabetes [86]. Tissue transglutaminase, in celiac disease, increases inflammation resulting in a reduction in the expression of the γ receptor (PPARG) activated by the proliferator peroxisomes, which may correlate with a reduced risk of developing type 2 diabetes [89,90]. According to the study by Tortora involving 98 patients with a newly diagnosed celiac disease, sevenpatients presented glucose levels indicative of hyperglycemia, whereas following 12 months of a gluten-free diet, hyperglycemia was reported in 25 patients (*p* < 0.01). The mean blood glucose value in the study group after a GFD increased from 86 mg/dL to 92 mg/dL [74]. In another study by Tortora, an increase in glucose levels was observed in patients with CD following a gluten-free diet in comparison with the baseline (mean ± SD: 88.7 ± 13.4 mg/dL vs. 84.1 ± 19.8 mg/dL) [75]. Moreover, as indicated by Zanini B et al., a statistically significant increase in fasting glucose after a gluten-free diet (12–48 months of observation) in 497 patients was observed—87.9 ± 10.8 vs. 89.7 ± 12.2 (*p* < 0.0001) [80].

### 3.6. Gluten-Free Diet and the Intestinal Microbiota

The gut microbiota affects the metabolic processes and immunity, consequently influencing the pathophysiological mechanisms of numerous diseases [91,92,93,94,95]. There is an increasing recognition supporting the hypothesis that alterations in both the composition and function of the intestinal microbiome are associated with many chronic inflammatory diseases, including celiac disease [96,97,98]. Nevertheless, research regarding the gut microbiome has certain limitations. Indeed, it usually involves small trials and the use of low-throughput techniques, such as culture techniques and simple molecular techniques which do not involve sequencing, thus they do not account for the entire gut microbiome [99]. In the study by Nistal E et al., the composition of the microbiome in healthy subjects (*n* = 11) was compared with the composition of the microbiome in patients with CD on a gluten-free diet (*n* = 11) and with patients suffering from CD who did not follow a GFD (*n* = 10). Differences in the composition of the microbiota involving *Lactobacillus* and *Bifidobacterium* strains (*p* < 0.05) between the healthy individuals and patients with CD were observed. Moreover, it was also reported that CD patients following a GFD presented a much higher concentration of short chain fatty acids (SCFA) in stools [94]. Additionally, in a study comparing the composition of microbiota in 24 CD patients (aged 2–12) not adhering to a GFD, 18 CD patients on a GFD (aged 1–12) for at least 2 years, and 20 healthy children not on a GFD (at the age of 2–11), it was found that *Bifidobacterium*, *Clostridium histolyticum, Clostridium lituseburense and Faecalibacterium prausnitzii* were less abundant (*p* < 0.050) in CD patients not on a GFD as compared to the control group. However, the ratio of the *Bacteroides-Prevotella* genera was more abundant (*p* < 0.050) in CD patients without a GFD than in the control group [100]. Another study involving children with celiac disease on a GFD (*n* = 19) and the healthy children (*n* = 15) found that the levels of *Lactobacillus, Enterococcus, and Bifidobacteria* were significantly higher (respectively: *p* = 0.028; *p* = 0.019; *p* = 0.023) in stool samples of the healthy individuals than in children with CD. In contrast, *Bacteroides*, *Staphylococcus*, *Salmonella*, *Shigella*, and *Klebsiella* were observed in significantly higher amounts (*p* = 0.014) in children with celiac disease than in the healthy subjects [101]. Furthermore, in the study involving 19 children (6–12 years old) complying with a GFD for at least 2 years and 15 healthy children, it was noted that differences in the biodiversity of the intestinal microbiota were found between the groups. Additionally, among the patients with CD, the differences were related to the celiac disease activity, and *Bacteroides vulgatus* and *Escherichia coli* were detected more frequently in this patient group than in the controls (*p* < 0.0001) [102]. Spanish research compared the microbiota composition of 20 children with a symptomatic CD who did not follow a GFD,10 children with asymptomatic CD who complied with a GFD for 1–2 years, and 8 healthy children who served as controls. The *Bacteroides* and *Escherichia* genera were more numerous in CD patients with active disease than in the control group, whereas the ratio of *Lactobacillus—Bifidobacterium* to *Bacteroides—Escherichia* was significantly reduced in patients with the active disease or the disease in remission compared to the control group [93]. According to the Italian study involving asymptomatic patients with CD (*n* = 7) on a GFD for a minimum of 2 years,patients with symptomatic CD (*n* = 7) who did not follow a GFD, and healthy controls (*n* = 7). It was observed that *Lactobacillus* abundance was lower in children with the active form of the disease compared to the healthy individuals and the patients on a GFD. Additionally, the characterization and differentiation of *Lactobacillus* strains in children with CD following a GFD was similar to that in healthy children. *Lactobacillus brevis*, *Lactobacillus rossiae* and *Lactobacillus pentosus* were identified only in faecal samples of healthy and asymptomatic patients with CD. In addition, *Lactobacillus fermentum*, *Lactobacillus delbrueckii subsp. bulgaricus* and *Lactobacillus gasseri* were identified in only a few stool samples of the healthy subjects. The ratio of *Lactobacillus* and *Bifidobacterium* to *Bacteroides* and *Enterobacteria* was lower in the group of children with CD on a GFD in comparison with the healthy children [103].

## 4. Conclusions

The research in the last decade has highlighted the fact that that an improperly balanced gluten-free diet might result in weight gain, thus, leading to obesity and the development of the metabolic disorders, such as type 2 diabetes, the metabolic syndrome and dyslipidemia. Therefore, a particular attention should be paid in order to maintain a balanced gluten-free diet in both newly diagnosed and ongoing celiac patients. In fact, there is little evidence to support the hypothesis that alterations in the gut microbiota leading to gut dysbiosis can be observed in patients with celiac disease treated with GFD.

However, it should be emphasized that GFD is the only effective method of treating celiac disease and should not be used without medical indications by healthy individuals, due to the risk of possible complications.

## Figures and Tables

**Figure 1 nutrients-13-00643-f001:**
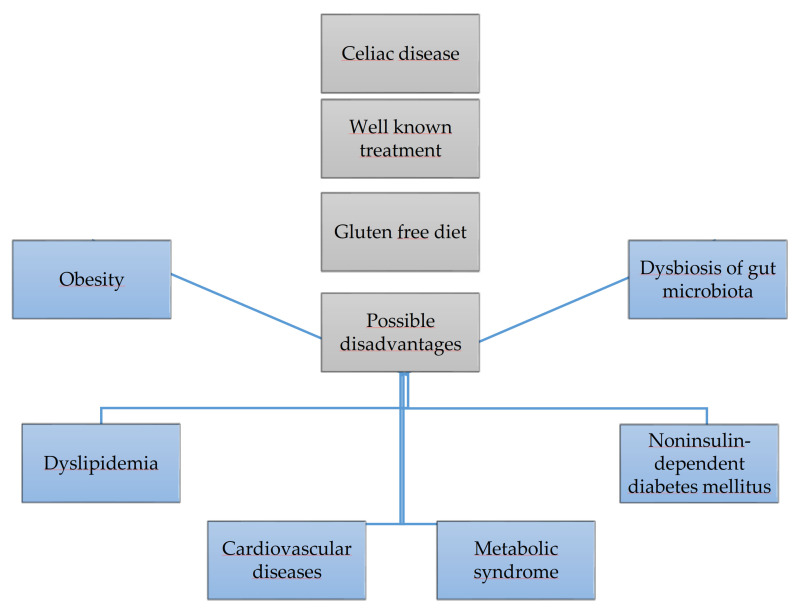
Gluten free diet and its risks.

**Table 1 nutrients-13-00643-t001:** Comparison of mean BMI values at the time of the diagnosis and after following a gluten-free diet.

Duration of GFD	N Cases	Mean BMI at Diagnosis	Mean BMI after GFD	*p*	References
12 months	98	22.9 (±4)	24.1 (±4)	0.01	[74]
56 months	370	23.2 (±3.6)	24.8 (±3.5)	<0.001	[75]
39.5 months	697	24.0	24.6	<0.001	[76]
at least a week	107	23.8 (21.5–27.5) *	25.1 (21.6–28.8) *	0.001	[77]
24.3 months (20.2–35.6 months) *	39	21.5 (20.4–25.1) *	22.6 (21.1–25.2)	0.07	[78]
18–24 months	78	45 **	78 **	<0.001	[79]
12–48 months	530	21.4 (±3.4)	22.5 (±3.5)	<0.0001	[80]
24 months	188	24.4	25.9	undefined	[69]
12 months	44	19.3 (±4.2)	20.9 (±4.1)	<0.001	[81]

BMI: Body Mass Index, GTD: Gluten-free diet *, median, ** patients with BMI > 25.

**Table 2 nutrients-13-00643-t002:** The lipid profile ofpatients at the time of the CD diagnosis and after the introduction of a gluten-free diet.

GFD Duration	N Cases	TC (mg/dL) before GFD	TC (mg/dL) after GFD	LDL (mg/dL) before GFD	LDL (mg/dL) after GFD	HDL (mg/dL) before GFD	HDL (mg/dL) after GFD	TG (mg/dL) before GFD	TG (mg/dL) after GFD	Reference
12 months	44	145.4 ± 38.7	158.0 ± 35.8 (*p* = 0.002)	88.6 ± 28.4	97.0 ± 24.0 (*p* = 0.001)	39.5 ± 10.3	42.8 ± 8.7	83 (38–360)	97 (53–436)	[81]
18–24 months	TC: 88 TG: 26 HDL:F: 70 M: 16	191.0 (90.0–322.0) *	228.0 (201.0–333.0) * (*p* = 0.0001)	-	-	F:48.0 (27.0–70.0) M: 52.0 (34.0–67.0)	F:35.0 (19.0–49.0) (*p* = 0.0001) M: 33.0 (26.0–39.0) (*p* = 0.0001)	127.0 (60.0–321.0) *	163.0 (151.0–332.0) * (*p* = 0.0006)	[79]
12–48 months	TC: 547 LDL, HDL: 72 TG: 522	171.2 ± 37.4	181.4 ± 35.1 (*p* < 0.0001)	102.5 ± 33.7	108.6 ± 34.0	48.1 ± 14.3	53.6 ± 14.2 (*p* < 0.0001)	87.9 ± 49.5	80.2 ± 42.8 (*p* < 0.0001)	[80]
Minimum 6 months median 20.5 months	132	169.2 ± 35.3	188.8 ± 2.8 (*p* < 0.0001)	109.7 ± 26.6	114.5 ± 28.1	45.8 ± 13.3	55.1 ± 13.6 (*p* < 0.0001)	79 (57–120) *	81 (63–118) *	[64]
56 months	370	163.7 ±35.2	179.4 ±29.3	117.2 ±39.1	119.8 ±38.1	54.4 ±13	54.2 ±12.8 (*p* = 0.003)	90.1 ±37.8	121 ±50.3	[75]

HDL—High-density lipoprotein, LDL—Low-density lipoprotein, TC – total cases, TG—Triglycerides, * median, M—male, F—female.

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
