# Peer review of "Multidimensional Disadvantages of a Gluten-Free Diet in Celiac Disease: A Narrative Review"

_nutrients, 2021, doi:10.3390/nu13020643_

Round 1

Reviewer 1 Report

A very interesting article on the disadvantages that a not well-balanced gluten-free diet may have on the overall health of the patients.

The paper is well executed. I have, however some queries:

I would add in the title: a narrative review, or otherwise, add a material and methods section and transform this paper into a systematic review. Given the amount of data analyzed in this paper, I think the latter would be a better solution.

In the introduction section (page 1, line 30,33) you should also incorporate this useful article: doi: 10.3390/medicina55090578.

I would delete figure 2, as I don't find it useful for the meaning of the paper.

Thank You

Author Response

Poznań, 05.02.2021

Martyna Marciniak
Department of Gastroenterology, Dietetics and Internal Diseases
Poznan University of Medical Science

Dear Reviewer,

On behalf of the authors' manuscript “Multidimensional disadvantages of a Gluten-Free Diet in Celiac Disease: comprehensive review”, I appreciate your helpful comments. We changed the title because it better shows the meaning of our manuscript.  We feel that the manuscript is now greatly improved. In accordance with the guidelines, we have introduced the following changes. All changes are marked in manuscript.

Reviewer 1.

A very interesting article on the disadvantages that a not well-balanced gluten-free diet may have on the overall health of the patients.

Reply:

Thank you for appreciating our work.

The paper is well executed. I have, however some queries:

I would add in the title: a narrative review, or otherwise, add a material and methods section and transform this paper into a systematic review. Given the amount of data analyzed in this paper, I think the latter would be a better solution.

Reply:

Thanks for this comment. In order to collect literature related to the presented topic, the PubMed database (www.pubmed.ncbi.nlm.nih.gov) was searched using the terms: “celiac disease”, “gluten-free diet”, “dyslipidemia”, “obesity”, “diabetes mellitus t.2”, “metabolic syndrome”, “cardiovascular diseases”, “gut microbiota”,  “microbiome gut dysbiosis”. As suggested, information on the methods used was added to the manuscript.

  1. Material and Methods

In order to collect literature related to the presented topic, the PubMed database (www.pubmed.ncbi.nlm.nih.gov) was searched using the terms: “celiac disease”, “gluten-free diet”, “dyslipidemia”, “obesity”, “diabetes mellitus t.2”, “metabolic syndrome”, “cardiovascular diseases”, “gut microbiota”,  “microbiome gut dysbiosis”.

We changed the title of our manuscript, we added a narrative review:Multidimensional disadvantages of a Gluten-Free Diet in Celiac Disease: a  narrative review”.

We changed the name of our manuscript to "narrative review", which is in line with the typology of Andrew Booth and colleagues (2012), which was developed on the basis of the so-called SALSA criteria (Search, AppraisaL, Synthesis and Analysis). The presented analysis of the literature covers the broadly understood topic of gluten-free diet at various levels of complexity. The search for studies included in the review was comprehensive. The summarizing method is narrative. The method of data analysis is chronological, conceptual and thematic.

In the introduction section (page 1, line 30,33) you should also incorporate this useful article: doi: 10.3390/medicina55090578.

Reply:

We added citation in the introduction section: 16. Abenavoli L, Dastoli S, Bennardo L, Boccuto L, Passante M, Silvestri M, Proietti I, Potenza C, Luzza F, Nisticò SP. The Skin in Celiac Disease Patients: The Other Side of the Coin. Medicina (Kaunas). 2019 Sep 9;55(9):578. doi: 10.3390/medicina55090578. PMID: 31505858; PMCID: PMC6780714.

I would delete figure 2, as I don't find it useful for the meaning of the paper.

Reply:

Thank  you for suggestion, the figure 2 was be deleted.

Yours faithfully,

Martyna Marciniak

Reviewer 2 Report

Current manuscript  entitled "Disadvantages of Gluten-Free Diet in Celiac Disease"  is an interesting review article, which deals with a rather interesting topic, with major concerns around the research community.

Here are some comments from the authors:

- Need extended editing for English language, please contact a certified English writer. 

- The title is very general and does not refer to what the text deals with, which are mainly those chronic conditions that are presented in Figure 1, (mainly metabolic diseases.)

- Inside the text there are some figures and tables for which there is no reference in the text. In the text, please  refer to every figure or table  by its number

- During the whole text, All References must be entered before the dot

- I strongly advise authors to read the excellent article entitled " Gluten-Free Diet in Celiac Disease — Forever and for All? by Itziinger et al. Nutrients 2018 and revise the whole text. 

Author Response

Poznań, 05.02.2021

Martyna Marciniak
Department of Gastroenterology, Dietetics and Internal Diseases
Poznan University of Medical Science

Dear Reviewer,

On behalf of the authors' manuscript “Multidimensional disadvantages of a Gluten-Free Diet in Celiac Disease: a narrative review”. We changed the title because it better shows the meaning of our manuscript. I appreciate your helpful comments.

We feel that the manuscript is now greatly improved. In accordance with the guidelines, we have introduced the following changes. All changes are marked in manuscript.

Reviewer 2.

Current manuscript  entitled "Disadvantages of Gluten-Free Diet in Celiac Disease"  is an interesting review article, which deals with a rather interesting topic, with major concerns around the research community.

Reply:

Thank you for appreciating our work.

Here are some comments from the authors:

- Need extended editing for English language, please contact a certified English writer. 

Reply:  

In the correction  of this manuscript we have cooperated with a biomedical translation company – TranslationLab – a certified English writer master Anna Zaborowska-CinciaÅ‚a, with whom we collaborated in the course of the publication of another paper (in Nutrients,  Journal of Clinical Medicine, International Journal of Molecular Sciences, Microorganism,  Obesity Reviews, Journal of Clinical Densitometry or Nutrition). All changes have been marked in the manuscript.

We would like to express our thanks for the cooperation, and we hope that our paper after corrections can undergo a further review process and be successfully published in Nutrients.

-The title is very general and does not refer to what the text deals with, which are mainly those chronic conditions that are presented in Figure 1, (mainly metabolic diseases.)

Reply:

Thank you for your comment, it is consistent with the suggestion of reviewer 1,  that‘s why we changed the title of the manuscript, which will highlight it as a narrative review, not only a glimpse at the metabolic diseases. The title changed“Multidimensional disadvantages of a Gluten-Free Diet in Celiac Disease: a  narrative review”.

- Inside the text there are some figures and tables for which there is no reference in the text. In the text, please  refer to every figure or table  by its number

 Reply:

Thank you for this comment.  The references of  figure and tables  was added in the text.

- During the whole text, All References must be entered before the dot

Reply:

Thank  you for this  comment. All References were entered before the dot.

- I strongly advise authors to read the excellent article entitled " Gluten-Free Diet in Celiac Disease — Forever and for All? by Itziinger et al. Nutrients 2018 and revise the whole text. 

Reply:

Thank  you for this  comment. We read this extremely interesting article: Itzlinger A, Branchi F, Elli L, Schumann M. Gluten-Free Diet in Celiac Disease-Forever and for All?. Nutrients. 2018;10(11):1796. Published 2018 Nov 18. doi:10.3390/nu10111796, and we strongly agree with the authors of this article, that it is advisable to follow a gluten-free diet, taking into account its numerous complications. We cited the recommended article [24] in the Introduction section, because it is a valuable confirmation that a gluten-free diet should be followed according to strictly  defined rules and recommendations.

Yours faithfully,

Martyna Marciniak

Round 2

Reviewer 1 Report

The authors responded to all queries. Paper is in my opinion publishable

Reviewer 2 Report

Manuscript is now really improved.  Congratulations